# Transcutaneous Auricular Vagus Nerve Stimulation Enhances Cerebrospinal Fluid Circulation and Restores Cognitive Function in the Rodent Model of Vascular Cognitive Impairment

**DOI:** 10.3390/cells11193019

**Published:** 2022-09-27

**Authors:** Seunghwan Choi, Dong Cheol Jang, Geehoon Chung, Sun Kwang Kim

**Affiliations:** 1Department of East-West Medicine, Graduate School, Kyung Hee University, 26 Kyungheedae-ro, Dongdaemun-gu, Seoul 02447, Korea; 2Department of Physiology, College of Korean Medicine, Kyung Hee University, 26 Kyungheedae-ro, Dongdaemun-gu, Seoul 02447, Korea

**Keywords:** vascular cognitive impairment, vagus nerve stimulation, cerebrospinal fluid

## Abstract

Vascular cognitive impairment (VCI) is a common sequela of cerebrovascular disorders. Although transcutaneous auricular vagus nerve stimulation (taVNS) has been considered a complementary treatment for various cognitive disorders, preclinical data on the effect of taVNS on VCI and its mechanism remain ambiguous. To measure cerebrospinal fluid (CSF) circulation during taVNS, we used in vivo two-photon microscopy with CSF and vasculature tracers. VCI was induced by transient bilateral common carotid artery occlusion (tBCCAO) surgery in mice. The animals underwent anesthesia, off-site stimulation, or taVNS for 20 min. Cognitive tests, including the novel object recognition and the Y-maze tests, were performed 24 h after the last treatment. The long-term treatment group received 6 days of treatment and was tested on day 7; the short-term treatment group received 2 days of treatment and was tested 3 days after tBCCAO surgery. CSF circulation increased remarkably in the taVNS group, but not in the anesthesia-control or off-site-stimulation-control groups. The cognitive impairment induced by tBCCAO was significantly restored after both long- and short-term taVNS. In terms of effects, both long- and short-term stimulations showed similar recovery effects. Our findings provide evidence that taVNS can facilitate CSF circulation and that repetitive taVNS can ameliorate VCI symptoms.

## 1. Introduction

Cerebrovascular disorders frequently result in vascular cognitive impairment (VCI), including post-stroke dementia, vascular dementia, and mild vascular cognitive impairment [1,2,3]. Symptoms of VCI are often progressive in stroke patients. In stroke, reduced blood flow triggers pathophysiological mechanisms, including hypoxia, inflammation, and oxidative stress, resulting in cerebral lesions [4]. Moreover, many patients experience recurrent strokes, which increases the prevalence of dementia symptoms. Despite its high prevalence, suitable treatments for improving VCI have not been well established. Treatments generally applied at present include Alzheimer’s disease drugs; however, only small benefits for VCI can be expected from these drugs [5]. Furthermore, the frequent side effects of conventional medications interfere with long-term dosing due to the progressive nature of the disease [6]. Stroke mainly occurs in the elderly and can be caused by other conditions, such as cerebrovascular embolic events, high-grade carotid artery stenotic plaque formation, and pregnancy-induced hypertension. For elderly or pregnant patients, the expected beneficial effects of chemical drugs often do not outweigh the concerns regarding side effects and fetal effects. Therefore, there is an unmet need for non-chemical, repeatable therapeutics for VCI management.

Vagus nerve stimulation (VNS) has been studied as an epilepsy treatment for decades [7,8,9]. Aside from its major role, numerous preclinical investigations have shown that VNS can improve cognitive function in animal models [10,11,12]. For example, VNS enhances memory retention in an animal model of fear conditioning [11,12]. The effects of VNS on cognitive function have also been studied in human clinical trials. In one report, a single VNS treatment improved the recognition memory in patients with seizures [13]. In another study, long-term treatment with VNS for 3–12 months improved the cognitive performance and prevented further decline in patients with Alzheimer’s disease (AD) [14,15].

Since the VNS device was initially developed as a surgically implanted device, most preclinical and clinical studies have focused on invasive VNS (iVNS) devices. Recent studies have reported that non-invasive VNS devices have similar therapeutic effects on neurological disorders, including cognitive impairment [10,16,17,18]. A recent report suggested that auricular transcutaneous VNS (taVNS), a noninvasive VNS, has a remarkable memory persistence effect in a fragile X syndrome mouse model [18]. Clinically, taVNS improves associative memory performance in healthy older individuals [17] and verbal order memory in healthy young individuals [16]. Nonetheless, no study has investigated the effects of taVNS on VCI.

The effects of VNS on the modulation of brain environment have not been fully elucidated. Recent animal studies have shown that VNS can promote the circulation of cerebrospinal fluid (CSF) in the brain. CSF plays an important role in cerebral homeostasis and metabolic waste clearance by exchanging water and biochemicals between arterial and venous blood and brain parenchyma [19,20,21]. CSF circulation is associated with cognitive function not only in mice [22] but also in human patients [23]. Although a recent study suggested that iVNS increases CSF circulation [24], it remains unclear as to whether taVNS induces a similar effect on CSF circulation. Furthermore, there is a lack of information regarding the changes in CSF circulation in vivo, as the previous study reported the changes only in vitro slice data.

In this study, we developed a novel taVNS device that allows for simultaneous stimulation not only to the cymba concha, a conventional taVNS site, but also to the surface of the dorsal auricle, a site where the vagus nerve branch innervates. Thus, we could stimulate a broad range of vagus nerve branches around the auricle. Using this device and the two-photon in vivo microscopy technique, we observed a considerable increase in CSF circulation during taVNS. We then investigated the effects of repetitive taVNS on cognitive function in a rodent model of stroke to determine whether these treatments could restore cognitive impairments due to cerebrovascular accidents.

## 2. Materials and Methods

### 2.1. Animals

All animal experiments were performed in accordance with the protocols approved by the Animal Care and Use Committee of Kyung Hee University (KHSASP-22-302). Eight-week-old male C57BL/6 mice were purchased from Daehan Bio Link (DBL, Umsung, Korea). The mice were housed in cages with food and water ad libitum. The room was maintained under a 12 h light/dark cycle and kept at 21 ± 1 °C. The animals were acclimatized for at least 2 weeks before an experimental schedule was created.

### 2.2. Electrode System

The electrode system was designed to fit and simultaneously stimulate the cymba concha and the surface of the dorsal auricle where the auricular branch of the vagus nerve (ABVN) innervates [25,26]. A 3D model made with the Shapr3D app was printed with a Form3 3D printer using gray resin. The device consisted of two parts: a frame for the stimulating electrode inserts and a neodymium magnetic electrode that delivered the electrical current transcutaneously (Figure 1A).

### 2.3. Stimulation Parameters

Electrical stimulation parameters were determined on the basis of previous studies showing cognitive-enhancing effects of taVNS [18,27]. Rectangular bipolar electrical pulses were transduced using the IX-RA-834 (iWorx Systems, Dover, NH, USA) data acquisition system. The total stimulation time of a single session was 20 min with a current of 1 mA, having a frequency of 20 Hz and a pulse width of 330 μs pulse width.

### 2.4. Procedures of Electrical Stimulation

The protocols for taVNS and off-site stimulation control were derived from an earlier study [18]. Stimulation was performed under isoflurane anesthesia (2% induction and 1.5% maintenance) in 0.8 L/min O_2_. Proper taVNS was executed on the cymba concha of the left ear. The off-site stimulation was delivered to the tip of the ear helix, which was innervated by the cervical spinal nerve but not by the ABVN (Figure 1C). To confirm that the proper stimulation was applied, an electrocardiogram (ECG) of lead II was recorded using an IX-BIO4-SA ECG device (iWorx Systems, Dover, NH, USA) connected to an IX-RA-834 data acquisition device. The recorded ECG signals were analyzed using Spike2 software (Cambridge Electronic Design, Cambridge, UK).

### 2.5. Transient Global Cerebral Ischemia

Surgery for inducing VCI was performed using previously described techniques [28]. Mice were anesthetized under isoflurane inhalation and were positioned supine over a heating blanket, with their body temperature maintained at approximately 36.5–37.5 °C. After making an incision on the ventral neck, the subcutaneous tissues and submandibular glands were retracted to expose the major arteries in the anterior cervical area. The common carotid arteries (CCA) were carefully dissected from the surrounding tissues, including the cervical vagus nerve. Aneurysmal clips were placed to occlude the bilateral CCA and were maintained for 25 min. After clip removal, the arteries were checked for reperfusion and absence of hemorrhage, and the surgical site was sutured. Sham procedures were performed with the same anesthesia and surgical protocols while keeping the CCAs intact. After the procedure, the animals were returned to a warming box that was kept at 30 °C.

### 2.6. Behavioral Assessments

All animals were randomly allocated to different experimental groups. The animals were assigned to either the long-term (6 days of stimulation, tested on day 7) or short-term (2 days of stimulation, tested on day 3) tests. All behavioral tests were performed in a separate room with a dim light and white noise and recorded with a video camera (HDR-CX405, Sony, Tokyo, Japan). Between each test session, the apparatus was thoroughly cleaned with 70% ethanol to remove olfactory cues.

### 2.7. Novel Object Recognition (NOR) Test

An arena with dimensions of 40 × 40 × 40 cm was used. In addition, blue cubes or red square-based pyramid plastic objects of 5 × 5 × 5 cm dimensions, which were similar in texture and volume, were used. The assessment scheme consisted of habituation and familiarization. The test phase was determined on the basis of previous studies [29,30,31]. The habituation phase of 10 min was given 1 day prior to the test phase in the empty open-field arena. In the familiarization phase, two identical objects were placed diagonally on the arena floor with Blu-tack mounting so that the animals could not move them. The objects were randomized and counterbalanced across animals. For the mice to explore the objects freely from all angles, a sufficient distance was adjusted between the walls and objects. After 10 min of the familiarization session, followed by a 1 h interval, a test session with two different objects was carried out. The animals were allowed to explore for 10 min, during which the time spent exploring each object was recorded. Object explorations were defined by the following criteria: the mouse was oriented toward the object, the snout was within 2 cm of the object, and the midpoint of the animal’s body was beyond 2 cm from the object. To calculate the percentage of total investigation time and the discrimination index, the following equations were used:$$\%\;of\;total\;investigation\;time=\frac{Investigation\;time\;of\;novel\;object}{Investigation\;time\;of\;novel\;object+Investigation\;time\;of\;familiar\;object}\times 100\;Discrimination\;index=\frac{Investigation\;time\;of\;novel\;object-Investigation\;time\;of\;familiar\;object}{Investigation\;time\;of\;novel\;object+Investigation\;time\;of\;familiar\;object}$$

### 2.8. Y-maze Test

A symmetrical Y-maze was used, and the test time was determined on the basis of previous studies [32,33]. During the 8 min test session, the mouse was placed in the center of the maze and allowed to explore the maze freely. After completing the session, the total number of entries and spontaneous alternations were calculated using a recorded video file. Spontaneous alternation, expressed as a percentage, was calculated using the following formula:$$Spontaneous\;alternation\;\left(\%\right)=\frac{number\;of\;spontaneous\;alternations}{total\;number\;of\;entries-2}\times 100$$

### 2.9. In Vivo Two-Photon Imaging

To implant cranial windows allowing for in vivo imaging of cortical CSF circulation, naive C57BL/6 mice were anesthetized with an intraperitoneal injection Zoletil (30 mg/kg) and xylazine (10 mg/kg) mixture. Following this, their body temperature was maintained at 37 °C using a heating pad [34,35]. Following scalp incision and removal of the connective tissue, a square-shaped craniotomy was performed on the site above the left cortex (0.5 mm posterior, 1.5 mm lateral to the bregma) without dura mater damage [36]. The exposed cortex was washed with artificial CSF (126 mM NaCl, 2.5 mM KCl, 1.25 mM NaH_2_PO_4_, 2 mM MgSO_4_, 2 mM CaCl_2_, 10 mM glucose, and 26 mM NaHCO_3_) and covered with a glass coverslip (2 × 2 mm; Matsunami Glass Ind., Ltd., Osaka, Japan). Using additional Vetbond and dental cement, a metal ring was implanted for head fixation in the microscope setup and fixed with the skull around the cranial window.

After mounting the cranial window, the cisterna magna was cannulated for infusion of the CSF tracer [37]. Each mouse was fixed in the stereotaxic frame; an incision was made over the posterior neck; and the neck muscles were retracted, thereby exposing the occipital crest and translucent dural membrane over the cerebellum and medulla. A 30 G dental needle cannula attached to a 5-cm PE10 tube filled with aCSF was inserted at an angle of 45° relative to the mouse’s head, which passed into the center of the cisterna magna. The cannula was inserted at a depth of 1–2 mm to avoid any damage to the cerebellum or medulla and was secured with Vetbond tissue adhesives and dental cement.

To visualize the CSF flow, 10 μL of FITC-d2000 (MW 2000 kD, Invitrogen, Waltham, WA, USA) dissolved in aCSF at a concentration of 0.5% was injected at a rate of 2 μL/min over 5 min with a syringe pump. For microsphere imaging, 10 μL of yellow-green polystyrene microspheres (FluoSpheres™ 1.0 µm, Invitrogen) dissolved in aCSF at a concentration of 0.5% was injected in the same manner. Then, 20 min of taVNS was applied before microsphere injection. A 0.1 mL bolus of 5% (*w*/*v*) Texas Red dextran (MW 70 kD, Invitrogen) in PBS was injected retro-orbitally for visualization of the cerebral vasculatures [35].

A Ti:sapphire laser system (Chameleon, Coherent, Santa Clara, CA, USA) attached to a two-photon microscope (FVMPE-RS, Olympus, Tokyo, Japan) was used for in vivo imaging. Two types of water immersion objective lens (XLPLN25XWMP2, 1.05 NA; LUMPLFLN40XW, 0.8 NA; Olympus, Tokyo, Japan) were used to image the cortex. The excitation wavelength was 870 nm for fluorescein isothiocyanate (FITC), Texas Red, and microspheres. Imaging was initiated 2 h after the injection of FITC or immediately after microspheres. Imaging planes that were 30–50 µm below the cortical surface were selected. Imaging of CSF flow in the cortex was conducted with a dual-channel (FITC and Texas Red, and microsphere and Texas Red) 512 × 512 pixel image acquisition. The dynamics of cortical CSF circulation following treatment were recorded with time-lapse imaging at 1 Hz every 5 min, including the time points before and after treatment. Microsphere images were acquired at 15 Hz for 5 min after the microsphere appeared in the imaging field. FITC images were processed and analyzed using ImageJ software (NIH, Bethesda, MD, USA), with the mean pixel intensity measured in each image at each time point. Cases with a mean intensity difference >5% from baseline at time point 0 were excluded. Microsphere particle tracking was conducted with the Trackmate imageJ plugin. Particles that moved at a total distance below 100 μm were excluded.

### 2.10. Data Analysis

All statistical analyses were performed using GraphPad Prism 9 and Microsoft Excel. One-way ANOVA with post hoc Dunnett’s or Tukey’s test, two-way repeated-measures ANOVA, and unpaired *t*-test were used. All graphs are shown as the mean ± SEM. The detailed statistical methods and numbers for each experiment are provided in the figure legends.

## 3. Results

### 3.1. Development of the Preclinical taVNS Device and the Experimental Setup

Preclinical devices for taVNS were manufactured using a 3D printing system. The device was designed to be mounted in the mouse auricular space using neodymium magnets and to provide simultaneous electrical stimulation on the cymba concha and surface of the dorsal auricle (Figure 1B). Animals were randomly assigned to the taVNS group or the off-site stimulation control group in a blinded manner, and the appropriate treatment was administered. In the taVNS group, electrical stimulation was administered under isoflurane anesthesia through a device that was placed on the cymba concha and dorsal auricle of the ear. Electrical stimulation was applied to the off-site stimulation control group in the same manner, except that the device was placed on both sides of the outer helix so that the vagus nerve branch was not directly stimulated (Figure 1C). To test the proper operation of the device and the specificity of the taVNS, we recorded electrocardiography data during the taVNS period and evaluated whether taVNS could alter the heart rate of the subjects, a reliable biomarker of VNS [38]. A significant reduction in heart rate was observed in the taVNS group, but not in the off-site stimulation control group (Figure 1D,E), validating the successful establishment of VNS and control stimulation. The taVNS group exhibited an approximately 15% reduction in heart rate after 20 min of stimulation, consistent with previous reports using taVNS or cervical VNS [39].

### 3.2. taVNS Promoted CSF Circulation

We investigated whether taVNS promotes CSF circulation in vivo. Impaired CSF circulation is critically involved in the development of VCI symptoms following ischemic injury [22] and is highly associated with cognitive impairment [23]. We examined CSF circulation in the cerebral cortex using in vivo two-photon microscopy with taVNS (Figure 2A). FITC-d2000 was injected into the cisterna magna to trace CSF circulation, and Texas Red dextran was injected intravenously to visualize blood vessels. In vivo imaging experiments were performed to track the spread of FITC-d2000 along the arterioles into the brain parenchyma (Figure 2B). The animals were randomly assigned to one of the following three groups in a blinded manner: the anesthesia control group, in which animals were anesthetized with isoflurane without electrical stimulation; the off-site stimulation control group, where animals were electrically stimulated but the device was placed on the ear helix so that the vagus nerve branch was not directly affected; or the taVNS group, where animals were stimulated at the proper VNS site (cymba concha and dorsal auricle). In the taVNS group, the normalized intensity of FITC-d2000, which indicates the amount of CSF circulation, increased significantly during the taVNS and continued to increase even after the stimulation was finished (Figure 2C). In contrast, the anesthesia control group and off-site stimulation control group showed no significant increase in FITC intensity (Figure 2C). For evaluating CSF dynamics in perivascular space after taVNS, we adopted fluorescent microsphere imaging. As the microsphere visualizes CSF moving, higher flow speed of microsphere movements in the perivascular spaces represents enhanced CSF circulation. After injecting microspheres into cisterna magna, microspheres appeared in the imaging field moving around the blood vessel (Figure 2D). Microsphere tracking showed that taVNS increased mean flow speed of microspheres (Figure 2E), representing taVNS-enhanced CSF flow in the perivascular space.

### 3.3. Transient Bilateral Common Carotid Artery Occlusion (tBCCAO) Induced Cognitive Impairment

To investigate VCI, a tBCCAO animal model of stroke was used. tBCCAO is a well-established experimental method that induces diffuse brain ischemic damage, resulting in cognitive impairment without motor deficits [28,40,41]. First, we validated whether cognitive impairment was induced by tBCCAO. The NOR and Y-maze tests were conducted 7 days after surgery (Figure 3A). There were no differences between groups in the total distance moved during the habituation period of the NOR test (Figure 3B). The percentages of total investigation time and the discrimination index scores evaluated by the NOR test were significantly lower in the tBCCAO groups compared to the sham group (Figure 3C,D). In the Y-maze test (Figure 3E), the tBCCAO groups showed significantly lower levels of spontaneous alternation than the sham group (Figure 3F). There was no difference in total arm entry, indicating that locomotion was intact after tBCCAO (Figure 3G). We then performed behavioral experiments in a new subset of animals 3 days after tBCCAO surgery (Figure 3H,I) and confirmed that the tBCCAO-induced cognitive impairment symptoms manifested during this early period (Figure 3J–N). Overall, these data indicate that tBCCAO caused cognitive impairment without affecting motor function, and the symptoms developed and persisted for at least 3~7 postoperative days.

### 3.4. Repetitive taVNS Restored Impaired Cognitive Function

Next, we evaluated the effects of taVNS on cognitive impairment induced by tBCCAO. Animals were randomly divided into three groups after stroke induction with tBCCAO: anesthesia control, off-site stimulation-control, and taVNS groups. Animals were subjected to each treatment for 6 days followed by the NOR and Y-maze tests on the seventh day (Figure 4A). Data from each group were compared to the data from the untreated tBCCAO group shown in Figure 3D,F. taVNS significantly ameliorated the discrimination index scores determined by the NOR test (Figure 4B) and the rates of spontaneous alternations in the Y-maze test (Figure 4C).

We then investigated whether the enhancement of cognitive function could be achieved by short-term taVNS in tBCCAO animals. A new subset of tBCCAO animals was subjected to each treatment of anesthesia, off-site stimulation, or taVNS for 2 consecutive days, and behavioral tests were performed on the third day (Figure 5A). Data from each group were compared to the data from the untreated tBCCAO group shown in Figure 3L,M. The discrimination index scores in the NOR test and rates of spontaneous alternations in the Y-maze test were significantly higher in the taVNS group than in the control group (Figure 5B,C).

To compare the size of the effect, the data measured on postoperative days 3 and 7 were pooled for each sham or tBCCAO group. Then, all behavioral data from each group were normalized by calculating the Z-score using the average value and standard deviation of the pooled sham group. Thus, the Z-score in this analysis indicates the relative cognitive function of the subjects in each group compared to the levels of normal subjects in the sham group. The tBCCAO group showed significantly lower levels of cognitive function than the sham group. The tBCCAO-induced impairment of cognitive function was significantly improved by either six or two repetitive taVNS treatments (Figure 6A,B). The degree of improvement was comparable between the two groups with six or two repetitive taVNS treatments. In addition, the data from the tBCCAO animals treated with taVNS were not significantly different from those of the animals subjected to the sham surgery. This indicates that the effect of either six or two repetitive taVNS treatments was sufficient to cancel out the tBCCAO-induced impairment of cognitive function.

## 4. Discussion

Clinically, VNS usually involves surgical implantation of a device onto the cervical branch of the vagus nerve. Therefore, VNS has been applied to limited conditions, such as epileptic seizures, despite its positive effects on various disorders. In this study, we developed a novel taVNS device on the basis of a magnet electrode that allows simultaneous stimulation of the vagus nerve branch in the cymba concha and the surface of the dorsal auricle. Although the device was designed for preclinical experiments in rodents, we believe that the same strategy can be utilized for clinical use.

We found that taVNS could facilitate the spread of the tracer including FITC dye and microsphere particles injected into the cisterna magna into the perivascular space, indicating enhanced CSF circulation. Previous studies have reported that VNS could enhance CSF circulation [24], and although the regulation of CSF circulation by VNS has been suggested, the studies were performed under slice conditions. Thus, in vivo evidence of the modulatory effects of VNS on CSF circulation is lacking. In addition, the effect of taVNS on CSF circulation has not been reported. To the best of our knowledge, our data provide the first in vivo evidence of the enhancement of taVNS-induced CSF circulation.

Impaired CSF circulation is closely associated with the manifestation of cognitive dysfunction. Clinical investigations have reported that patients with cognitive dysfunction show impairments in CSF circulation and that the extent of impairment is correlated with the severity of symptoms [23]. In this study, we demonstrated for the first time that taVNS can improve cognitive function in a rodent model of VCI. Notably, behavioral assessments were performed 24 h after taVNS treatment in our study. Thus, improvement in cognitive function was not mediated by direct short-term activation of the vagus nerve. Rather, we suggest that repetitive taVNS exerted a long-lasting effect via the CSF-mediated clearance of harmful metabolites from tissues.

The interaction between CSF circulation and post-stroke symptoms has been studied for several decades. Recent studies have shown that aberrant CSF circulation after stroke is mediated by glymphatic system dysfunction [42]. CSF circulation in the interstitial space, non-polarized astrocytic aquaporin-4, and inflammatory responses of the perivascular space are critically involved in pathological changes [42,43,44]. After a stroke insult, brain edema occurs through cytotoxic, ionic, and vasogenic mechanisms following the time course of ischemia [45]. Loss of glymphatic function due to an ischemic attack contributes to brain edema, causing secondary injury [46,47]. In addition, toxic metabolites, such as amyloid-beta and Nogo-A, which are produced during the progression of ischemic stroke and get deposited in the brain parenchyma, could result in secondary degeneration with neuronal death and poor neurological outcomes with memory function impairment [48,49,50]. In this respect, restoring glymphatic function would help to flush out fluids and toxic solutes from the site of injury and prevent secondary brain damage. For the future direction, investigating the CSF-containing metabolites and inflammatory factors before and after taVNS may be interesting. Moreover, the effect of cholinergic modulation to the CSF circulation could be an intriguing study since VNS enhances cholinergic release [51].

A notable point in our results is that short-term taVNS exerted a sufficient effect to ameliorate VCI symptoms. Initially, we tested the effect of a long-term treatment in model animals involving taVNS six times, considering that the glymphatic system could be persistently impaired even 7 days after a stroke attack [52]. We speculated that repetitive and chronic intervention in the glymphatic system would be required to ameliorate cognitive impairment symptoms [18]. Nevertheless, our experimental results showed that short-term treatment with repetitive taVNS twice was sufficient to improve cognitive function. The relatively immediate effect of taVNS, along with its high efficiency and safety, emphasizes the great potential of this treatment for clinical application.

## Figures and Tables

**Figure 1 cells-11-03019-f001:**
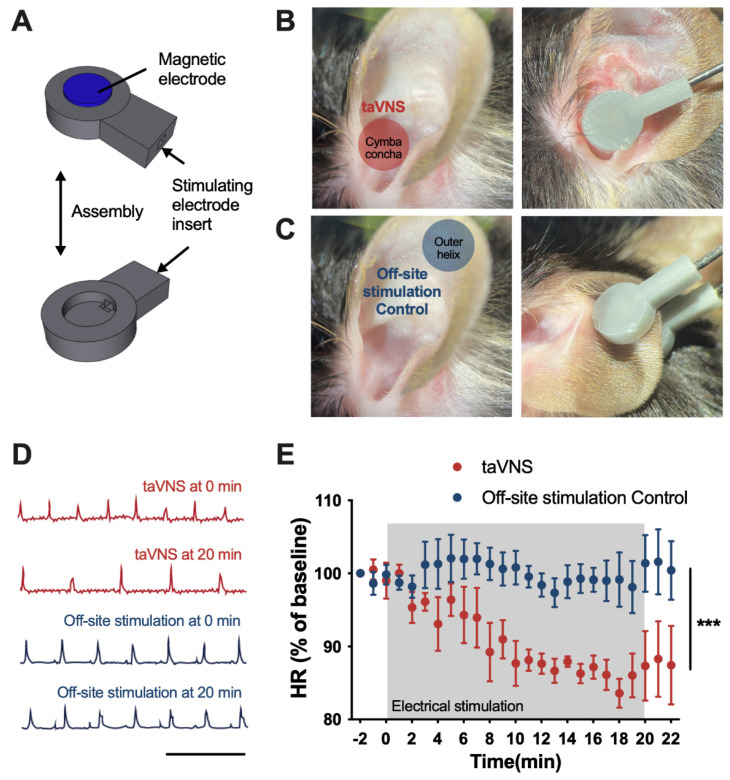
Novel device for auricular vagus nerve stimulation and modulatory effects on heart rates. (**A**) Three-dimensional modeling of stimulation devices. (**B**,**C**) Three-dimensionally printed prototype device applied to the cymba concha (taVNS) and outer helix (off-site stimulation control). (**D**) Representative ECG trace showing the declined heart rate in a model mouse, corresponding to prolongation of the R-R interval during taVNS. Scale bar 500 ms. (**E**) taVNS suppressed heart rate; however, there were no changes in the off-site stimulation control group (taVNS, n = 5; off-site stimulation control, n = 6; *p* < 0.001). The error bars indicate the SEM. Two-way repeated measures ANOVA was used. *** *p* < 0.001.

**Figure 2 cells-11-03019-f002:**
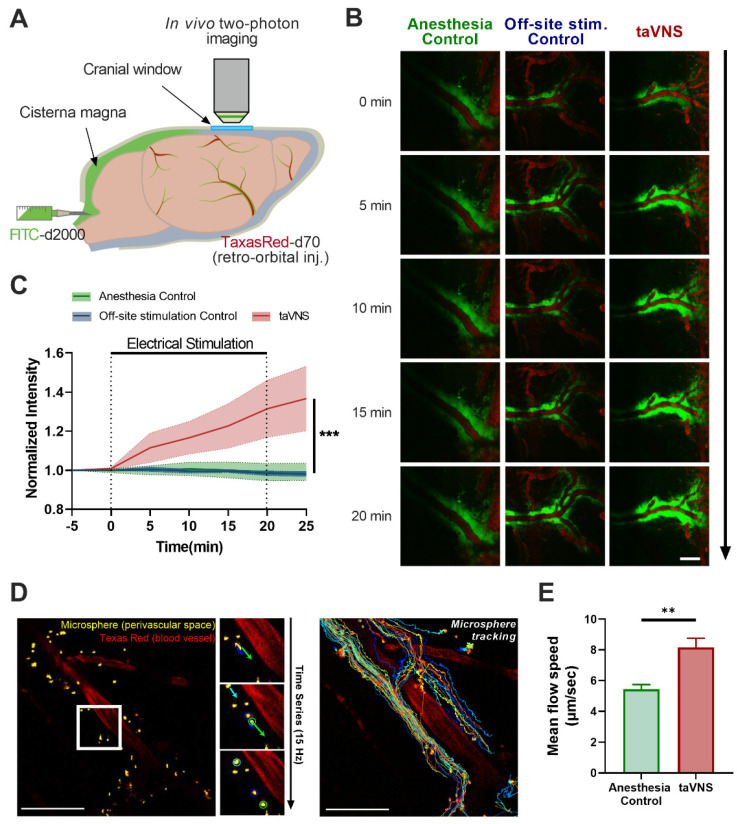
taVNS-enhanced cortical CSF circulation. (**A**) Schematic image showing CSF tracer (FITC-dextran) injection via cisterna magna cannulation and in vivo two-photon imaging of cortical CSF dynamics. (**B**) Representative images of the cortical CSF tracer after 20 min of stimulation. Scale bar: 100 μm. (**C**) taVNS (n = 5) promoted CSF tracer fluorescence intensity compared to the off-site stimulation control (n = 5, *p* < 0.001) and anesthesia (n = 6, *p* < 0.001) groups. (**D**) Representative images showing microspheres moving in perivascular space and microsphere particle tracking analysis. Scale bar: 50 μm. (**E**) Mean flow speed of microspheres increased in the taVNS group (n = 29) compared to anesthesia control (n = 14) each from 2 mice. The error bars indicate the SEM. Two-way repeated-measures analysis of variance (ANOVA) and unpaired *t*-test were used. ** *p* < 0.01; *** *p* < 0.001.

**Figure 3 cells-11-03019-f003:**
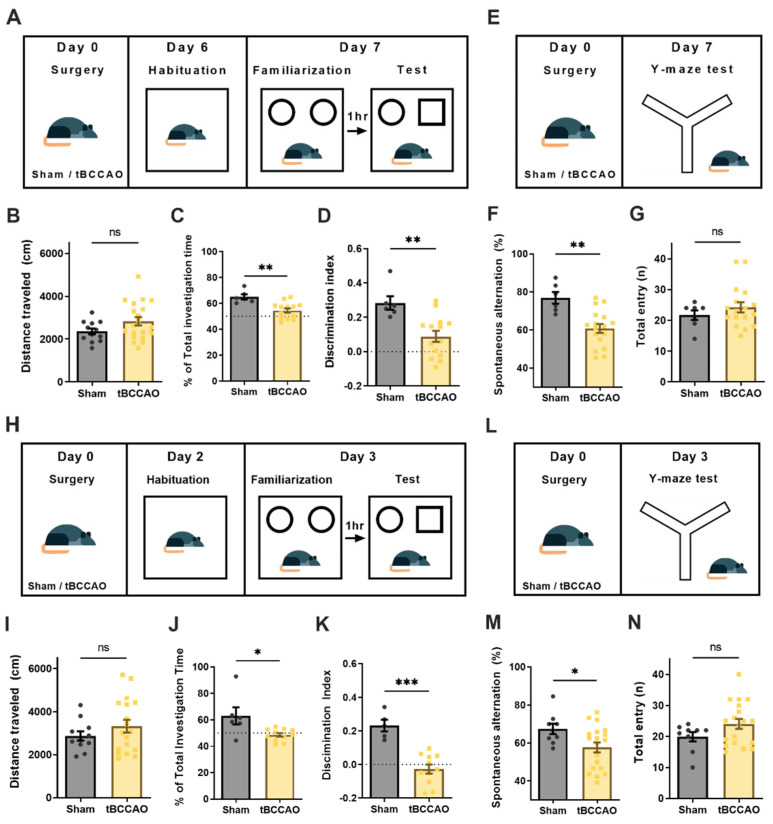
Cognitive impairments induced by transient bilateral common carotid artery occlusion (tBCCAO). Recognition memory and short-term working memory were evaluated using the novel object recognition test (NOR) and Y-maze test, respectively, after 7 days and 3 days. (**A**) Experimental scheme of the NOR test after 7 days. (**B**) Locomotion evaluated by travel distance in the open area was not affected by tBCCAO (sham, n = 6; tBCCAO, n = 14). (**C**) The rate of total investigation time was significantly decreased after tBCCAO (*p =* 0.001). (**D**) The discrimination index score was significantly reduced in the tBCCAO group (*p =* 0.003). (**E**) Experimental scheme of the Y-maze test after 7 days. (**F**) Spontaneous alternation rates were considerably lower in the tBCCAO group (*p =* 0.001; sham, n = 6; tBCCAO, n = 17). (**G**) The rate of total arm entry was not changed. (**H**) Experimental scheme of the NOR test after 3 days. (**I**) tBCCAO did not affect the locomotion after 3 days (sham, n = 6; tBCCAO, n = 11). (**J**) The percent of total investigation time decreased 3 days after tBCCAO (*p =* 0.013). (**K**) The discrimination index decreased significantly following tBCCAO (*p* < 0.001). (**L**) Experimental scheme of the Y-maze test after 3 days. (**M**) Spontaneous alternation rate was significantly damaged (*p* = 0.032; sham, n = 9; tBCCAO, n = 19). (**N**) The number of total entries was not influenced. The error bars indicate the SEM. An unpaired *t*-test was used. * *p* < 0.05; ** *p* < 0.01; *** *p* < 0.001; ns, not significant.

**Figure 4 cells-11-03019-f004:**
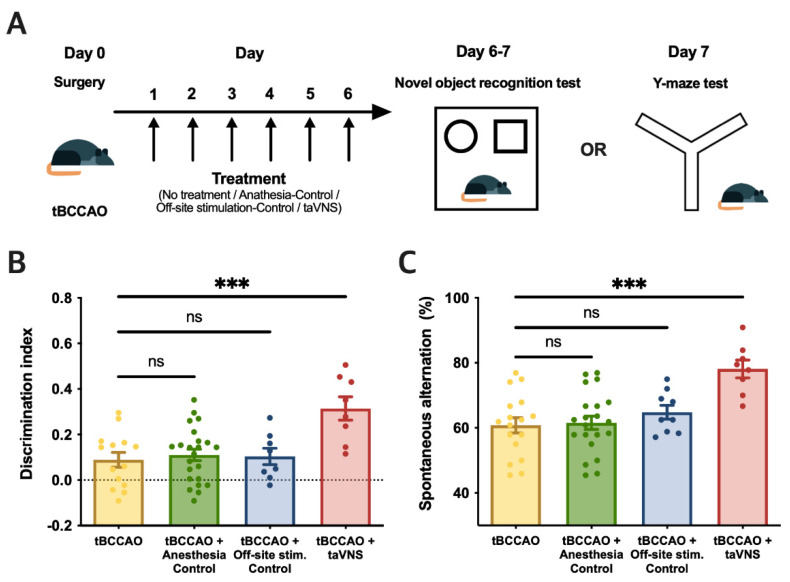
Impaired cognitive function was significantly restored after 6 phases of taVNS. (**A**) Experimental scheme. (**B**) In the NOR test, the discrimination index score of the taVNS treatment group (red, n = 8) significantly increased compared to the tBCCAO group (no treatment, yellow, n = 14; *p* < 0.001). The discrimination index scores of both the anesthesia control group (green, n = 23) and off-site stimulation control group (blue, n = 8) were not different from the tBCCAO group. (**C**) In the Y-maze test, the spontaneous alternation rate of the taVNS treatment group (red, n = 8) was significantly higher than that of the tBCCAO group (no treatment, yellow, n = 17; *p* < 0.001). The spontaneous alternation rates of both the anesthesia control group (green, n = 21) and off-site stimulation control group (blue, n = 9) were not different from the tBCCAO group. The error bars indicate the SEM. One-way ANOVA with the post hoc Dunnett test was used. *** *p* < 0.001; ns, not significant.

**Figure 5 cells-11-03019-f005:**
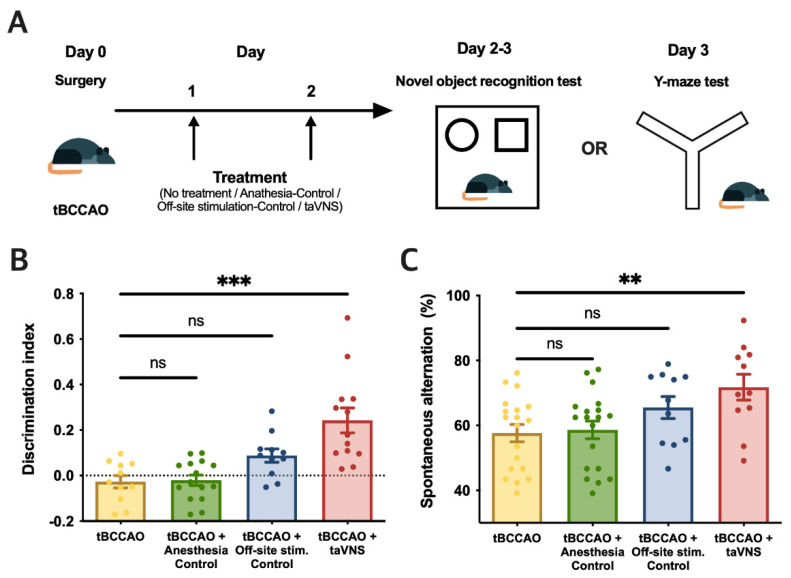
Impaired cognitive function was also significantly restored after two taVNS sessions. (**A**) Experimental scheme. (**B**) In the NOR test, the discrimination index score of the taVNS treatment group (red, n = 13) significantly increased in comparison to the tBCCAO group (no treatment, yellow, n = 11; *p* < 0.001). The discrimination index scores of both the anesthesia control group (green, n = 15) and off-site stimulation control group (blue, n = 11) were not different from the tBCCAO group. (**C**) In the Y-maze test, the spontaneous alternation rate of the taVNS treatment group (red, n = 11) was significantly higher than that of the tBCCAO group (no treatment, yellow, n = 19; *p =* 0.008). The spontaneous alternation rates of both the anesthesia control group (green, n = 19) and off-site stimulation control group (blue, n = 11) were not different from the tBCCAO group. The error bars indicate the SEM. One-way ANOVA with post hoc Dunnett test was used. ** *p* < 0.01; *** *p* < 0.001; ns, not significant.

**Figure 6 cells-11-03019-f006:**
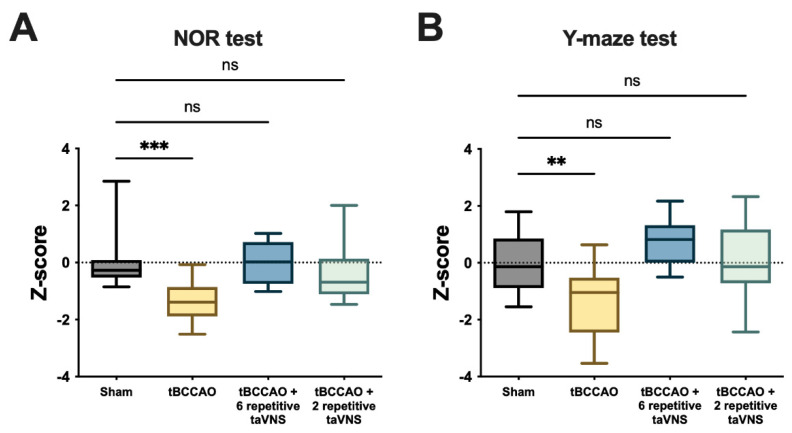
The effect sizes of both six and two repetitive days of treatment were similar. Z−score representing the size of the effect, normalized by the average and standard deviation of the sham group, was adopted to compare each group. (**A**) In the NOR test, the group that received six repetitions of taVNS (deep blue, n = 8) and the group that received two repetitions of taVNS (light blue, n = 13) showed similar Z−scores to that of the sham group (gray, n = 12). The tBCCAO group had a significantly lower Z−score of tBCCAO (yellow, n = 25; *p* < 0.001). (**B**) Similar to the NOR test results, in the Y-maze test, the group that received six repetitions of taVNS (deep blue, n = 8) and the group that received two repetitions of taVNS (light blue, n = 11) showed similar Z−scores to that of the sham group (gray, n = 36). The tBCCAO group had a significantly lower Z−score of tBCCAO (yellow, n = 15; *p* = 0.001). The error bars indicate the SEM. One−way repeated measures ANOVA with the post hoc Dunnett test was used. ** *p* < 0.01; *** *p* < 0.001; ns, not significant.

## Data Availability

The data that support the findings of this study are available from the corresponding author upon reasonable request.

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
