# Peer review of "Transcutaneous Auricular Vagus Nerve Stimulation Enhances Cerebrospinal Fluid Circulation and Restores Cognitive Function in the Rodent Model of Vascular Cognitive Impairment"

_cells, 2022, doi:10.3390/cells11193019_

Round 1

Reviewer 1 Report

This manuscript "Transcutaneous auricular vagus nerve stimulation enhances cerebrospinal fluid circulation and restores cognitive function in the rodent model of vascular cognitive impairment" by Choi et al. deals with the very interesting topic of non invasive brain stimulation, literature in taVNS is increasing and taVNS a promising tool in treating various neurological and psychiatric but also immunological an many other diseases. Less side effects and the easy use are big advantages.

I have two major which are listed below.

Major comments: 

1. In the introduction, last paragraph, the author mention that they have developed a new stimulation paradigm including two stimulation sides: the cymba concha and the surface of the dorsal auricle. The author should comment on this and argue why this is important. Moreover, the author used only one set of stimulation parameters. The author should comment which this stimulation parameters were used though the paradigm has been changed.

2. The data seems of good quality, the experimental setup is complicated and elaborate. The statistics is in principle sufficient for this kind of study. However, very different size of groups are compared in this study, though the variability in each group seems relatively large, e.g. n=8 vs 17, etc. 

The authors should add data in order to adjust sample groups which are compared.

Author Response

This manuscript "Transcutaneous auricular vagus nerve stimulation enhances cerebrospinal fluid circulation and restores cognitive function in the rodent model of vascular cognitive impairment" by Choi et al. deals with the very interesting topic of non invasive brain stimulation, literature in taVNS is increasing and taVNS a promising tool in treating various neurological and psychiatric but also immunological an many other diseases. Less side effects and the easy use are big advantages.

We greatly appreciate the reviewer for the thoughtful comments and important suggestions. We did our best to be responsive to them.

I have two major which are listed below.

Major comments:

1. In the introduction, last paragraph, the author mention that they have developed a new stimulation paradigm including two stimulation sides: the cymba concha and the surface of the dorsal auricle. The author should comment on this and argue why this is important. Moreover, the author used only one set of stimulation parameters. The author should comment which this stimulation parameters were used though the paradigm has been changed.

  • In the revised manuscript, we supplemented a brief comment regarding the stimulation sites. Now the sentence reads as follows:
    “In this study, we developed a novel taVNS device that allows simultaneous stimulation not only to the cymba concha, a conventional taVNS site, but also to the surface of the dorsal auricle, a site where the vagus nerve branch innervates. Thus, we could stimulate a broad range of vagus nerve branches around the auricle.”
  • We determined the stimulation parameters according to the previous study of cognitive impairment [1]. The study showed taVNS improved memory persistence in naive and intellectual disability mice. To clarify the parameter determination, we changed the sentence in the Materials & Methods section below:
    “Electrical stimulation parameters were determined based on previous studies showing cognitive-enhancing effects of taVNS”

2. The data seems of good quality, the experimental setup is complicated and elaborate. The statistics is in principle sufficient for this kind of study. However, very different size of groups are compared in this study, though the variability in each group seems relatively large, e.g. n=8 vs 17, etc. The authors should add data in order to adjust sample groups which are compared.

  • As the reviewer’s concern, different sample sizes of groups are able to cause statistical bias, which is usually because of homogeneity of variances between the groups. Thus, we ran Levene’s test to evaluate the homogeneity of variances of data presented in Figure 4,5 and 6. All data passed the criteria of Levene’s test. We found our data meet the statistical requirement and there is no statistical bias.

  1. Vázquez-Oliver, A.; Brambilla-Pisoni, C.; Domingo-Gainza, M.; Maldonado, R.; Ivorra, A.; Ozaita, A. Auricular Transcutaneous Vagus Nerve Stimulation Improves Memory Persistence in Naïve Mice and in an Intellectual Disability Mouse Model. Brain Stimul 2020, 13, 494–498, doi:10.1016/j.brs.2019.12.024.

Reviewer 2 Report

The manuscript describes auricular vagus nerve stimulation to enhance cerebrospinal fluid circulation to restore restore impaired vascular cognitive function in a mouse model. The topic is of interest and first describes a new technical approach with relevance for possible clinical application. 

however, some minor points should be clarified: 

1) In the introduction page 1; line 40 the causes for stroke should mention cerebrovascular embolic events, high-grade carotid artery stenotic plaque formation as more specific causes of stroke.  

2) For induction of transient global cerebral ischemia a 25 min occlusion of both CCA was performed. How did the authors find this particular time for CCA occlusion, are the ischemic changes all  transient or did the authors observe any  microscopic changes or infarctions ? 

3) In figures 1E and 2 C, please show the lines in the figures more clearly for the different groups investigated using interrupted lines, dotted lines or round points, squared points or other symbols.    4) In the discussion, page 11, line 369, the sentence should start with "Impared CSF circulation is closely associated with the manifestation of cognitive dysfunction.

5) Are these vagal effects mediated by receptor activation, most likel ymuscarinic receptor activation. Meaning that cholinergic stimulation using agonists would show the same results increasing CSF circulation or, are the effects seen with taVNS stimulation blocked by atropine or specific anatgonists ?                      

Author Response

The manuscript describes auricular vagus nerve stimulation to enhance cerebrospinal fluid circulation to restore restore impaired vascular cognitive function in a mouse model. The topic is of interest and first describes a new technical approach with relevance for possible clinical application.

however, some minor points should be clarified:

We appreciate the reviewer’s careful comments. We did our best to respond to these valuable comments.

1) In the introduction page 1; line 40 the causes for stroke should mention cerebrovascular embolic events, high-grade carotid artery stenotic plaque formation as more specific causes of stroke. 

  • In the revised manuscript, we added the causes for stroke as the reviewer’s suggestion. Now the sentence reads as follows:
    “Stroke mainly occurs in the elderly and can be caused by other conditions, such as cerebrovascular embolic events, high-grade carotid artery stenotic plaque formation and pregnancy-induced hypertension.”

2) For induction of transient global cerebral ischemia a 25 min occlusion of both CCA was performed. How did the authors find this particular time for CCA occlusion, are the ischemic changes all  transient or did the authors observe any  microscopic changes or infarctions ?

  • We adopted the particular time point of 25 minutes according to the previous report [1] (reference number 28) that used the same cognitive impairment model caused by transient global ischemia with tBCCAO. In the respect of occlusion time, Soares et al. tried various times of 12, 17, and 25 minutes. Although the occlusion was transient in all the cases, the ischemic damage induced the infarctions in the brain areas. They reported that the infarction could be observed in the hippocampus, shown by the presence of shrunken neurons as opposed to well-preserved neurons observed in sham control. For example, 25 minutes of tBCCAO resulted in damage to the hippocampal region. The numbers of intact neurons were less than half the sham-operated control level at 7, 14, and 28 days after tBCCAO.

3) In figures 1E and 2 C, please show the lines in the figures more clearly for the different groups investigated using interrupted lines, dotted lines or round points, squared points or other symbols. 

  • We have changed symbols in figure 1E and colors in figure 2C in the revised manuscript.

 4) In the discussion, page 11, line 369, the sentence should start with "Impared CSF circulation is closely associated with the manifestation of cognitive dysfunction.

  • In the revised manuscript, we edited the sentence as the reviewer’s suggestion.

5) Are these vagal effects mediated by receptor activation, most likely muscarinic receptor activation. Meaning that cholinergic stimulation using agonists would show the same results increasing CSF circulation or, are the effects seen with taVNS stimulation blocked by atropine or specific anatgonists ?

  • Since the current manuscript is the first taVNS study on vascular cognitive impairment, we are preparing a separate mechanism study. We are considering the mechanism of taVNS in the aspect of cholinergic stimulation as reviewer’s suggestion. As we will consider the reviewer’s comment during our follow up study, we added sentences in Discussion as follows:
    “Moreover, the effect of cholinergic modulation to the CSF circulation could be an intriguing study since VNS enhances cholinergic release.”

  1.  Soares, L.M.; Schiavon, A.P.; Milani, H.; Oliveira, R.M.W. de Cognitive Impairment and Persistent Anxiety-Related Responses Following Bilateral Common Carotid Artery Occlusion in Mice. Behav Brain Res 2013, 249, 28–37, doi:10.1016/j.bbr.2013.04.010.

Reviewer 3 Report

In this study, Choi et al., proposes that transcutaneous auricular vagus nerve stimulation (taVNS) enhances CSF circulation and restores cognitive function in mice. This is an interesting approach, with the potential to be developed into a therapeutic treatment for humans. Thus, the study is important. However, the data does not conclusively demonstrates that taVNS enhances CSF circulation.

The authors show that following taVNS, cognitive function is restored in tBCCAO mice. However, it seems that the conclusions are merely based on the fluorescence data presented in Fig 2. To further support their hypothesis, additional experiments are needed. For example, can the authors show that taVNS increases CSF circulation in normal mice? Such as with fluorescence beads or some other approach?

To further support the claim that taVNS leads to increased CSF clearance, it would be important to show that the approach does indeed causes a decrease in harmful metabolites as stated in their Discussion. Can the authors measure inflammatory markers in the CSF of tBCCAO treated mice?

How is taVNS-induced decreases in heart rate associated with better glymphatics? What impact does this approach has on vessel pulsatility?

The behavioral test were performed in different animal groups. Can these behavioral tests be performed in the same animal before and after taVNS? With and without tBCCAO?

Author Response

In this study, Choi et al., proposes that transcutaneous auricular vagus nerve stimulation (taVNS) enhances CSF circulation and restores cognitive function in mice. This is an interesting approach, with the potential to be developed into a therapeutic treatment for humans. Thus, the study is important. However, the data does not conclusively demonstrates that taVNS enhances CSF circulation.

The authors show that following taVNS, cognitive function is restored in tBCCAO mice. However, it seems that the conclusions are merely based on the fluorescence data presented in Fig 2. To further support their hypothesis, additional experiments are needed. For example, can the authors show that taVNS increases CSF circulation in normal mice? Such as with fluorescence beads or some other approach?

  • Fluorescent CSF tracers have been accepted as a universal method for CSF imaging. Although the results of ex vivo studies are accepted [1], our in vivo data could explain CSF circulation by taVNS more clearly. As the reviewer points out, measuring dynamics of microspheres (or fluorescence beads) in flowing perivascular spaces could be utilized [2]. However, even Mestre et al. pointed out that infused microspheres could aggregate or adhere to the perivascular spaces wall, making the imaging field messy.

To further support the claim that taVNS leads to increased CSF clearance, it would be important to show that the approach does indeed causes a decrease in harmful metabolites as stated in their Discussion. Can the authors measure inflammatory markers in the CSF of tBCCAO treated mice?

  • Recently, one study showed that the level of several inflammatory markers in the CSF are elevated after tBCCAO [3]. It may be obvious that tBCCAO elevates the levels of inflammatory markers, but it is an interesting question whether taVNS reduces inflammatory markers or harmful metabolites as reviewer mentioned. Since we are trying to solve this question separately from the current study as our first follow up study, we added the sentence in Discussion as follows:
    “For the further direction, investigating the CSF-containing metabolites and inflammatory factors before and after taVNS may be interesting.”

How is taVNS-induced decreases in heart rate associated with better glymphatics? What impact does this approach has on vessel pulsatility?

  • As we mentioned in the Results line 220-223, we recorded the heart rate to confirm whether our stimulation was properly delivered, because the reduction of heart rate is a reliable biomarker of successful taVNS.
  • Although we did not intend to associate the decreased heart rate following vagus nerve activation with the changes in the glymphatic system, we found this question quite interesting. Glymphatic circulation and arterial hemodynamics are closely related in that CSF influx is driven by arterial pulsatility. It is plausible that the reduced heart rate caused by taVNS and ensuing changes in the blood pressure affected the pulsatility of blood vessels. Indeed, a previous study reported that the upregulation in the blood pressure resulted in the decreased flow of cerebrospinal fluid [2]. And, increased glymphatic influx is correlated with low heart rate [4]. According to the studies so far, the relationship between the glymphatic circulation and the VNS is attributed to the regulation of cardiac factors by the VNS.

The behavioral test were performed in different animal groups. Can these behavioral tests be performed in the same animal before and after taVNS? With and without tBCCAO?

  • The behavioral test we have performed in the current study is not repeatable. Since both novel object recognition (NOR) test and Y-maze test depend on memory of the mouse, the previous trial obviously affects the follow up trial. Thus, separating animal groups was the only way to compare with less experimental bias.

  1. Cheng, K.P.; Brodnick, S.K.; Blanz, S.L.; Zeng, W.; Kegel, J.; Pisaniello, J.A.; Ness, J.P.; Ross, E.; Nicolai, E.N.; Settell, M.L.; et al. Clinically-Derived Vagus Nerve Stimulation Enhances Cerebrospinal Fluid Penetrance. Brain Stimul 2020, 13, 1024–1030, doi:10.1016/j.brs.2020.03.012.
  2. Mestre, H.; Tithof, J.; Du, T.; Song, W.; Peng, W.; Sweeney, A.M.; Olveda, G.; Thomas, J.H.; Nedergaard, M.; Kelley, D.H. Flow of Cerebrospinal Fluid Is Driven by Arterial Pulsations and Is Reduced in Hypertension. Nat Commun 2018, 9, 4878, doi:10.1038/s41467-018-07318-3.
  3. Yan, F.; Tian, Y.; Huang, Y.; Wang, Q.; Liu, P.; Wang, N.; Zhao, F.; Zhong, L.; Hui, W.; Luo, Y. Xi-Xian-Tong-Shuan Capsule Alleviates Vascular Cognitive Impairment in Chronic Cerebral Hypoperfusion Rats by Promoting White Matter Repair, Reducing Neuronal Loss, and Inhibiting the Expression of pro-Inflammatory Factors. Biomed Pharmacother 2022, 145, 112453, doi:10.1016/j.biopha.2021.112453.
  4. Hablitz, L.M.; Vinitsky, H.S.; Sun, Q.; Stæger, F.F.; Sigurdsson, B.; Mortensen, K.N.; Lilius, T.O.; Nedergaard, M. Increased Glymphatic Influx Is Correlated with High EEG Delta Power and Low Heart Rate in Mice under Anesthesia. Sci Adv 2019, 5, eaav5447, doi:10.1126/sciadv.aav5447.

Round 2

Reviewer 1 Report

My two concerns have been adressed, no further comments.

Author Response

Thank you for your sincere peer comments.

Reviewer 3 Report

While the authors made some changes to the text, the main conclusions for this study (as defined in the title of the article) are based on the data presented in one figure. To fully support the idea that taVNS enhances CSF circulation, additional parameters need to be included in the article.

Author Response

As reviewer 3 pointed out that additional data are necessary to support transcutaneous auricular vagus nerve stimulation (taVNS) enhances CSF circulation, we conducted fluorescent microsphere imaging.

In the result of microsphere particle tracking, mean flow speed was increased after taVNS compared to the anesthesia control. It suggests taVNS could facilitate CSF circulation, in the same context as FITC dye spreading well with taVNS. The supplemented results are shown in Figure 2, which is attached in PDF format.

We would like to thank your sincere comments that are helpful to our research.
